# Immunogenicity and Safety of Pneumococcal Vaccines Co-Administered with Common Travel Vaccines in Adults: A Systematic Review [note 1]

**DOI:** 10.3390/vaccines13060643

**Published:** 2025-06-14

**Authors:** Raziyeh Niyati, Omid Rezahosseini, Christina Ekenberg, Carsten Schade Larsen, Zitta Barrella Harboe

**Affiliations:** 1Department of Pulmonary and Infectious Diseases, Copenhagen University Hospital, North Zealand, 3400 Hillerød, Denmark; raziyeh.niyati@regionh.dk (R.N.); christina.ekenberg.01@regionh.dk (C.E.); 2Department of Infectious Diseases, Aarhus University Hospital, 8200 Aarhus, Denmark; carslars@rm.dk; 3Department of Clinical Medicine, Faculty of Health and Medical Science, University of Copenhagen, 2200 København, Denmark

**Keywords:** immunogenicity, safety, pneumococcal, diphtheria, tetanus, pertussis, Hepatitis A, Hepatitis B, meningococcal, travel, vaccine

## Abstract

**Background:** Co-administration of vaccines can impact the immune response and safety. We aim to systematically review the current scientific literature and find evidence regarding the immunogenicity and safety of pneumococcal vaccines co-administered with common vaccines that are recommended for travelers, including hepatitis A, hepatitis B, yellow fever, tetanus, diphtheria, and acellular pertussis (Tdap), Japanese encephalitis, rabies, typhoid, or meningococcal (MCV) vaccine in adults (18 years or older). **Methods:** We followed the PRISMA 2020 guidelines and used the PICOS process to select the keywords. We searched PubMed, Web of Science, Scopus, EMBASE, and Google from 1 January 2000 to 30 June 2024. We included randomized controlled trials, non-randomized controlled trials, observational studies, case series, and case reports in adults, all published in English. **Results:** Out of 598 articles screened, 6 studies were included in our study. Three studies involved immunocompetent individuals, and three involved immunocompromised individuals. Co-administration of pneumococcal vaccine with Tdap or Hepatitis A in immunocompetent individuals was safe and immunogenic. Similar findings were reported for immunocompromised individuals when pneumococcal vaccines were co-administered with Tdap, hepatitis A, and hepatitis B. However, no reports investigated the co-administration of yellow fever, rabies, Japanese encephalitis, and typhoid. Two non-randomized studies in immunocompromised individuals had a high risk of bias. **Conclusions:** The studies collectively indicate that the co-administration of pneumococcal vaccines with Hepatitis A and Tdap vaccines in adult immunocompetent and immunocompromised individuals is safe and immunogenic. However, a knowledge gap remains, and further high-quality studies are needed, particularly due to the limited number of studies and the potential risk of bias.

## 1. Introduction

Vaccination is an evidence-based, effective preventive strategy against many infectious diseases and is widely recommended to individuals who travel to endemic areas where there is a risk of acquiring potentially severe infections [1]. Therefore, susceptible travelers are recommended or required to be vaccinated against certain vaccine-preventable infections (VPIs), including pneumococcal, yellow fever, hepatitis A, and hepatitis B, among others. Vaccine recommendations are based on age, vaccination history, and immune status of the traveler, as well as the endemicity of disease at the destination and travel duration [2,3].

A pre-travel consultation can be an opportunity to apply preventive strategies for travelers, particularly as the mean age of intercontinental travelers continues to rise and an increasing number of older adults are seeking pre-travel consultations [4,5,6]. Older adults are susceptible to infections such as pneumococcal disease, and therefore, the pneumococcal vaccines are particularly relevant for these individuals, as they often have underlying medical conditions that further elevate their risk of severe infections and complications [7]. In this context, understanding the safety and immunogenicity of co-administering pneumococcal vaccines with commonly recommended travel vaccines is critical to optimizing travel health consultations.

Vaccine co-administration, defined as administering two or more vaccines in the same visit, is a routine clinical practice aiming to increase vaccination coverage [8]. However, our knowledge of the immunogenicity or safety of the co-administered vaccines in adults is limited since most evidence comes from studies in children [8]. The gap in evidence in adults is especially important because immune responses in adults can differ from those in children due to several factors, including immunosenescence, prior antigen exposure, and the higher prevalence of chronic comorbidities [9]. These age-related variations can influence both the magnitude and durability of vaccine-induced protection, suggesting that findings from pediatric studies cannot be directly extrapolated to adult populations. In addition, co-administration of vaccines can result in potential immunological interactions such as carrier-induced epitopic suppression (CIES), bystander interference, and carrier-specific enhancement of T-Cell Help, which can potentially affect immune response to vaccines [10]. Therefore, we conducted a systematic review to answer whether the co-administration of pneumococcal vaccines with common travel vaccines in adults impacts the immunogenicity and safety of the administered pneumococcal vaccine in adults. The findings can help to decrease knowledge gaps, inform healthcare practitioners, and refine vaccination schedules.

## 2. Materials and Methods

### 2.1. Search Strategy

We conducted this systematic review in accordance with the Preferred Reporting Items for Systematic Reviews and Meta-analyses (PRISMA 2020) guidelines [11]. We aimed to answer the clinical question: “Does the co-administration of pneumococcal vaccines with common travel vaccines in adults reduce the immunogenicity of pneumococcal vaccines?”.

Common travel vaccines were hepatitis A, hepatitis B, yellow fever, diphtheria-tetanus booster, Japanese encephalitis, rabies, typhoid, or meningococcal vaccines. SARS-CoV-2, Influenza, and Herpes Zoster vaccines are also among the vaccines that are recommended for adult vulnerable travelers. However, we have investigated the co-administration of pneumococcal vaccines with these vaccines in two previous systematic reviews and meta-analyses, and therefore, SARS-CoV-2, Influenza, and Herpes Zoster were excluded from the current work [12,13].

We utilized the PICOS (Population, Intervention, Comparator, Outcome, and Study) framework. To gather relevant studies, we extensively searched multiple databases, including PubMed, Web of Science, Scopus, EMBASE, and MEDLINE, covering the period from 1 January 2000 to June 2024. Two independent reviewers (RN and OR) screened titles and abstracts using predefined search terms to assess relevance. We then conducted full-text reviews of potentially eligible articles based on predetermined inclusion criteria. A third reviewer (ZBH) resolved any discrepancies to ensure consistency in the selection process. Additionally, we examined the reference lists of included studies and performed a manual search on Google to identify any relevant studies not found in the initial database search. The protocol was registered in the Open Science Framework (OSF) (osf.io/fw7qz). A summary of this study was presented at ESCMID Global 2025 [14].

### 2.2. Inclusion Criteria

We included studies in which a pneumococcal vaccine was co-administered with hepatitis A, hepatitis B, yellow fever, diphtheria-tetanus, Japanese encephalitis, rabies, typhoid, or a meningococcal vaccine in adults (18 years or older). We included randomized controlled trials, as well as non-randomized controlled trials, observational studies, case series, and case reports reporting information from adults. Furthermore, only human studies published in English were considered for inclusion. We excluded editorials, commentaries, opinions, review articles, and conference proceedings if they did not contain original data.

### 2.3. Full Search Strategy

We used a combination of Medical Subject Headings (MeSHs) and free-text terms to search the PubMed/MEDLINE databases. In PubMed, the MeSH-based search strategy yielded 138 records: ((((((“Pneumococcal Vaccines” [Mesh]) OR (“10-valent pneumococcal conjugate vaccine” [Supplementary Concept] OR “five-valent pneumococcal conjugate vaccine” [Supplementary Concept] OR “Heptavalent Pneumococcal Conjugate Vaccine” [Mesh])) OR “13-valent pneumococcal vaccine” [Supplementary Concept]) OR “23-valent pneumococcal capsular polysaccharide vaccine” [Supplementary Concept]) OR “PHiD-CV vaccine” [Supplementary Concept]) AND ((((((((((((“hepatitis A Vaccines” [Mesh]) OR (“hepatitis B Vaccines” [Mesh] OR “hepatitis B vaccine AS04” [Supplementary Concept])) OR “Engerix-B” [Supplementary Concept]) OR “Recombivax HB” [Supplementary Concept]) OR “Heplisav-B” [Supplementary Concept]) OR “Yellow Fever Vaccine” [Mesh]) OR “Diphtheria-Tetanus Vaccine” [Mesh]) OR (“Diphtheria-Tetanus-acellular Pertussis Vaccines” [Mesh] OR “diphtheria-tetanus-acellular pertussis-Hib-hepatitis B vaccine” [Supplementary Concept] OR “diphtheria-tetanus-five component acellular pertussis-inactivated poliomyelitis-*Haemophilus influenzae* type b conjugate vaccine” [Supplementary Concept])) OR “Japanese Encephalitis Vaccines” [Mesh]) OR “Rabies Vaccines” [Mesh]) OR (“Typhoid-Paratyphoid Vaccines” [Mesh] OR “typhoid vaccine M01ZH09” [Supplementary Concept] OR “Ty21a typhoid vaccine” [Supplementary Concept])) OR “Meningococcal Vaccines” [Mesh])) AND (((((((((((“Immunogenicity, Vaccine” [Mesh]) OR “Immunity” [Mesh]) OR “Antibody Formation” [Mesh]) OR “Seroconversion” [Mesh]) OR “Vaccine Efficacy” [Mesh]) OR “Immunologic Memory” [Mesh]) OR “Antibodies, Neutralizing” [Mesh]) OR “Immunity, Humoral” [Mesh]) OR “Immunity, Cellular” [Mesh]) OR “adverse effects” [Subheading]) OR “Safety” [Mesh]).

For the free-text search in PubMed, we applied the following combination of search terms, which resulted in 238 records: ((“Immunogenicity” OR “Adverse Effects” OR “Safety” OR “Reactogenicity” OR “Side Effects” OR “Immune Response” OR “Antibody Response” OR “Seroconversion” OR “Vaccine Efficacy” OR “Immunologic Memory” OR “Immunoprotection” OR “Neutralizing Antibodies” OR “T-cell Response” OR “Humoral Immunity” OR “Cell-mediated Immunity”) AND (“Pneumococcal Vaccines” OR “Pneumococcal Conjugate Vaccine” OR PCV OR “Pneumococcal Polysaccharide Vaccine” OR PPV OR Prevnar OR Synflorix OR Pneumovax) AND (“hepatitis A Vaccine” OR Havrix OR Vaqta OR Avaxim OR “hepatitis B Vaccine” OR Engerix-B OR Recombivax HB OR Heplisav-B OR “Yellow Fever Vaccine” OR YF-VAX OR Stamaril OR “Diphtheria-Tetanus Vaccine” OR DTaP OR DT OR Td OR Infanrix OR Daptacel OR Adacel OR Boostrix OR “Japanese Encephalitis Vaccine” OR Ixiaro OR JENVAC OR JE-VAX OR “Rabies Vaccine” OR Imovax Rabies OR RabAvert OR Rabipur OR “Typhoid Vaccine” OR Typhim Vi OR Vivotif OR Typherix OR “Meningococcal Vaccine” OR Menactra OR Menveo OR Menomune OR Nimenrix OR Bexsero OR Trumenba)) AND (“co-administration” OR “co-administration” OR “simultaneous co-administration “concurrent” OR “combined” OR “concomitant”). The same but adjusted combination of free-text terms was used to search EMBASE, Scopus, and Web of Science. All search results were imported into Covidence (https://www.covidence.org/), which is a widely used web-based platform designed to streamline systematic review processes. Covidence facilitates key review stages, including duplicate removal, screening, study selection, data extraction, and risk of bias assessment. The platform also supports team collaboration by enabling transparent and coordinated decision-making throughout the review.

### 2.4. Risk of Bias Assessment

We evaluated the risk of bias using the Cochrane Risk of Bias 2 (RoB 2) [15] for randomized clinical trials and the Risk of Bias in Non-randomized Studies of Interventions (ROBINS-I) [16]. To visualize the assessments, we used the Robvis tool, which generates traffic light plots to depict the risk of bias of the included studies [15]. These risk of bias tools are standard instruments that apply structured signaling questions and predefined algorithms to identify potential sources of bias within studies. They assess multiple domains, including study methodology, participant selection, data collection, outcome reporting, and statistical analysis.

By systematically identifying methodological weaknesses, these tools help determine the reliability and validity of study findings. Our assessments categorized the risk of bias as low, some concerns, or high for randomized trials, and as low, moderate, or high for non-randomized studies.

### 2.5. Synthesis of Results

Due to the heterogeneity in inclusion criteria, populations, and vaccine combinations across the studies, performing a meta-analysis was not possible. Therefore, the results were reported narratively.

## 3. Results

We used broad search term combinations, removed duplicates, and 598 articles were available for title and abstract screening. However, considering the aim of the systematic review and the inclusion criteria, only 15 studies were eligible for full-text review. Finally, six studies were included, comprising four randomized controlled trials and two retrospective cohort studies (Figure 1). Three of the six studies involved vaccinating individuals from the general population (immunocompetent), while three included individuals with a known immunosuppressive state (Table 1). Two of the latter studies were after bone marrow transplantation, and one investigated patients with rheumatoid arthritis undergoing treatment. The studies investigated the co-administration of pneumococcal vaccines with hepatitis A, B, meningococcal, and diphtheria-tetanus boosters. Furthermore, we could not find any information regarding the co-administration of pneumococcal and yellow fever, Japanese encephalitis, rabies, and typhoid vaccines.

### 3.1. Studies in Immunocompetent Adults

In a phase IV randomized controlled trial by Riekkinen et al. [17], 304 healthy adult volunteers from Sweden and Finland were included. Individuals with a known history of immunosuppression or prior pneumococcal or hepatitis A vaccination were not included in this study. The participants were divided into three groups. Group one received pneumococcal conjugate vaccine 13-valent (PCV13, Prevnar13^®^) plus hepatitis A (Havrix^®^, or Epaxal^®^), the second group received PCV13 alone, and the third group received the hepatitis A vaccine alone. Post-vaccination IgG and opsonophagocytosis assay (OPA) were measured 28 days after vaccination, and values were generally similar across the groups, except for a higher OPA for pneumococcal serotype 4, and a higher IgG titer for serotype 3 in the concomitant vaccination group [17]. The subanalysis on older participants showed statistically significantly lower OPA in 12 of the 13 investigated serotypes (except for serotype 19A) and a lower IgG response for six (serotypes 1, 3, 6A, 6B, 9V, and 19F) of the 13 serotypes. No immediate adverse effect was observed; however, on day 28 after vaccination, 80% of the participants in concomitant or single PCV13 administration groups equally reported at least one adverse effect. Contrary to what was expected, there was a trend toward higher proportions of local adverse events or fever after a single PCV13 administration than in co-administration; however, this finding was not statistically significant [17].

Tashani et al. [18] performed an RCT to investigate the safety and immunogenicity of PCV13 co-administered with tetanus, diphtheria, and acellular pertussis (Tdap) or quadrivalent meningococcal conjugate vaccine (MCV4). The study included 111 participants aged 18–64 from Australian travelers before attending the Hajj pilgrimage [18]. The participants were randomized into three groups. Group A received Tdap (Boostrix^®^) first, followed 3–4 weeks later by PCV13 (Prevnar13^®^) and MCV4 (Menveo^®^). Group B received Tdap, PCV13, and MCV4 concomitantly on the same day, and group C was vaccinated with PCV13 and MCV4 first, followed 3–4 weeks later by Tdap. The immune response was measured 3–4 weeks after vaccination. The study showed that administering the Tdap vaccine 3–4 weeks prior to the concurrent administration of PCV13 and MCV4 significantly impaired the immune response to six of the 13 pneumococcal serotypes [18]. Comparison of geometric mean titers (GMTs) across the three study groups revealed that Group A had significantly lower GMTs for six serotypes (3, 5, 18C, 4, 19A, and 9V) than Group C. The six other serotypes (19F, 1, 14, 23F, 7F, and 6A) also had a lower but statistically insignificant GMT value in Group A than in Group C. However, the differences were not statistically significant [18]. No statistically significant differences were found between Groups B and C across all serotypes, although GMTs in Group B were slightly lower than in Group C for most serotypes. Immune response to serotypes 6B and 6A consistently ranked among the top four serotypes with the highest OPA GMTs across all three groups, while serotypes 3, 5, 1, and 19F had the lowest immune response [18]. The highest GMT recorded for a serotype was 6B in Group C, and the lowest was serotype 3 in Group A. Most local and systemic adverse events within seven days of vaccination were mild to moderate. Local adverse events, swelling, redness, and pain were similar among study groups but were least common when vaccines were administered separately. The most frequent systemic reactions were drowsiness and muscle aches, with Tdap causing the fewest systemic adverse events, followed by the combination of PCV13 and MCV4. No serious adverse event occurred, and all reactions resolved within seven days [18]. Tashani et al. concluded that in adults receiving multiple vaccines, such as those preparing for travel, the PCV13 vaccine is best administered either concurrently with or prior to the Tdap vaccine, rather than following it [18]. However, a significant limitation of this study is that the study does not include a non-concomitant group for PCV13, and no PCV13 + Tdap or PCV13 + MCV4 administration arm for comparison.

Beyond this, we did not find information on the timing of vaccine co-administration in the studies included in this systematic review.

The study by Song et al. [19] aimed to investigate the safety and immunogenicity of co-administration of a tetanus-diphtheria (Td) vaccine with PCV13 in 462 adults older than 50 years with stable underlying diseases for at least six weeks. The study excluded individuals with a history of pneumococcal infection in the past five years, prior pneumococcal vaccination, Td vaccination within the previous ten years, known immunodeficiencies, or use of immunosuppressive therapy. The participants were included in the RCT from 2013 to 2016 in Korea and divided into three groups: Group 1 received both Td (Td-pur^®^) and PCV13 (Prevnar13^®^), Group 2 received PCV13, and Group 3 received Td alone. The baseline OPA GMTs for the four serotypes (1, 5, 18C, and 19A) were similar between Group 1 (Td + PCV13) and Group 2 (PCV13 alone). The immune response was measured four weeks after vaccination. After PCV13 vaccination, OPA titers increased significantly for all serotypes in both groups, with all subjects reaching an OPA titer of ≥8.

The study met the non-inferiority criteria for GMT ratios across all four evaluated serotypes following co-administration. Overall, post-vaccination OPA GMTs were comparable between groups. Notably, serotype 1 elicited a significantly higher OPA GMT in Group 1 (PCV13 + Td) compared to Group 2 (PCV13 alone). A key limitation of the study was that only 4 of the 13 pneumococcal serotypes were assessed. Local reactions did not differ significantly across the three groups, regardless of whether vaccines were given together. The most commonly reported local reaction was injection site pain, often with tenderness: 63.6% in Group 1, 56.4% in Group 2, and 54.1% in Group 3. Systemic reactions such as fatigue, myalgia, and arthralgia occurred more frequently in participants receiving PCV13 (Groups 1 and 2) than in those receiving Td alone (Group 3). No serious vaccine-related adverse events were observed [19].

### 3.2. Studies in Immunocompromised Patients

Winthrop et al. [20] conducted an RCT to investigate pneumococcal (Prevnar 13^®^) and Tdap (Boostrix^®^) vaccination in 106 patients with rheumatoid arthritis receiving baricitinib. Baricitinib is an oral Janus kinase (JAK 1/2) inhibitor, which decreases inflammation by blocking the activity of JAK 1/2 [21]. The participants were enrolled from the USA and Puerto Rico. Adults (18 years or older) with moderately to severely active rheumatoid arthritis and those on a stable dose of methotrexate (MTX) or corticosteroids for more than six weeks before entry were included. The exclusion criteria were as follows: previous vaccination with 23-valent Pneumococcal Polysaccharide Vaccine (PPSV-23) in the last 12 months or PCV13 at any time, prior tetanus vaccination within the last five years, recent positive history of pneumococcal infection, or other infections requiring hospitalization, or those on cyclosporine, leflunomide, or azathioprine. In the baricitinib-treated group, the geometric mean concentrations (GMCs) of anti-pneumococcal IgG antibodies increased from baseline (0.48–1.04 μg/mL) to 5 weeks post-vaccination (1.39–5.84 μg/mL) with statistically significant elevations for all serotypes at weeks 5 and 12. Although antibody concentrations slightly decreased from week 5 to week 12, they remained significantly higher than baseline. Serotypes 1, 18c, and 5 had the highest, while serotypes 4 and 3 had the lowest IgG GMCs. The geometric fold rise (GMFR) responses ranged from two to five folds for the 13 serotypes at weeks 5 and 12. Serotypes 1, 5, and 18 had the highest, while serotypes 4, 14, and 3 had the lowest GMFRs. The study did not include a control group that received a single PCV13 vaccination, which is a limitation of the study. Thirty (28%) adverse events were reported during the vaccine follow-ups. Seven patients (6.6%) reported injection-site events, which included pain and erythema; one patient reported moderate pain at the PCV13 site. Three patients (2.8%) experienced severe adverse events, but these events were considered not related to vaccine administration by the authors [20].

Sattler et al. [22] included 84 adult patients in Germany after allogeneic hematopoietic stem cell transplantation (HSCT). The patients underwent allogenic HSCT and survived at least nine months without relapse. Additionally, participants had received at least three consecutive vaccinations and had not been administered intravenous immunoglobulin within the three months preceding or following vaccination. The PCV13 (Prevnar13^®^) was administered to 55 participants. The pneumococcal 10-valent conjugate vaccine (PCV10, Synflorix^®^) was administered in 12 HSCT recipients concomitantly with vaccines against five other infections. The concomitant vaccines were against diphtheria (D), tetanus (T), Bordetella pertussis (aP), *Haemophilus influenzae* type b (Hib), and inactivated poliomyelitis virus (IPV). In this study, the following combination vaccines were administered: Pentavac^®^ (DTaP-IPV-Hib), Infanrix hexa/IPV-Hib^®^ (DTaP-hepatitis B-IPV-Hib), or Hexyon^®^ (DTaP-hepatitis B-IPV-Hib), and the median time from vaccination to serum collection was 124 days (range from 21 to 1350 days) in those who received pneumococcal vaccines.

The immune response against pneumococcal vaccines was measured in 67 participants, and 56 participants (83.6%) achieved protective antibody levels. No statistically significant differences in patients’ characteristics and medical history were found when comparing patients who achieved protective titers and those without positive titer response [22]. Unfortunately, the study did not include a control group that received a single pneumococcal vaccination.

In a retrospective cohort study, Palazzo et al. [23] investigated the safety and immunogenicity of the standard inactivated vaccine series in a group of patients with multiple myeloma (MM) who received autologous hematopoietic stem cell transplantation (aHSCT) and were on lenalidomide maintenance therapy. Lenalidomide is an anti-tumor and immunomodulatory agent and a derivative of thalidomide used for the treatment of MM [24]. The participants were included from 2010 to 2014 in the USA and began their immunizations one year post-transplant. MM patients who underwent their first aHSCT within one year of diagnosis and did not receive a tandem transplant were included. The participants were vaccinated with PCV13 (Prevnar13^®^), Hib conjugate vaccine (ActHib^®^), poliovirus inactivated vaccine (IPOL^®^), Tdap (Boostrix^®^, or Adacel^®^), hepatitis A and B vaccine (Twinrix^®^, or Havrix^®^), and measles, mumps, and rubella vaccine.

The vaccination schedule incorporated combination vaccines where possible. These vaccines were typically administered during the same visit in three doses at intervals of 1 to 3 months. None of the participants were immune according to their antibody titers before vaccination. Pneumococcal vaccination was completed in 119 of 120 patients (99%). Among 119 patients, 118 received the Hib vaccine, 109 received the Polio vaccine, 106 received the Tdap vaccine, 56 received the hepatitis A vaccine, and 68 received the hepatitis B vaccine concomitantly. Response to pneumococcal vaccines was demonstrated in 69 patients (58%). The lack of a control group receiving a single administration of PCV13 is a limitation of this study. There was no statistical difference in the response rate for patients receiving LM versus those who did not receive this agent. No vaccine-related adverse events were reported in this study.

### 3.3. Risk of Bias

Three of the four RCTs had a low risk of bias, and one had some concerns in the randomization process (Figure 2). However, both non-randomized studies had a serious risk of bias, with specific concerns about missing data and confounding factors (Figure 3).

## 4. Discussion

We found six heterogeneous studies that co-administered pneumococcal vaccines with vaccines against hepatitis A, hepatitis B, meningococcal, diphtheria, tetanus, pertussis, poliomyelitis, and MMR in adults. Overall, the studies showed that immune responses to pneumococcal vaccines across most pneumococcal serotypes are not statistically significantly affected by co-administered vaccines. However, some variations have been reported depending on the type of co-administered vaccines, as well as the participant’s age and immune status. Furthermore, no serious adverse effect was reported after concomitant administration of the vaccines. We could not find information about the co-administration of PCV with yellow fever, rabies, typhoid, and Japanese encephalitis vaccines. We speculate that the lack of information on this topic might stem from the fact that these vaccines are commonly co-administered during travel preparations, and there may be a general trust in existing guidelines. As a result, no dedicated studies have been performed to investigate this further.

The serotype-specific antibody responses were affected differentially by the co-administration of vaccines. Of note, the study by Riekkinen et al. showed that concomitant administration of PCV13 and hepatitis A vaccine is not only non-inferior to single administration of PCV13 but also is positively associated with the immune response to serotypes 4 and 3 [17]. Both serotypes 3 and 4 are among the common causes of invasive pneumococcal infections in adults [25]. Serotype 3 is recognized as one of the most problematic pneumococcal serotypes, characterized by poor or non-antibody responses after vaccination and causing severe infections such as empyema, meningitis, and sepsis [26]. Therefore, a strategy that can improve the immune response against this serotype is of clinical value and can be investigated in future studies.

The time interval between vaccinations is another factor that can affect the immune response [27]. The study by Tashani et al. showed that the co-administration of PCV13 with Tdap and MCV4 vaccines does not decrease the immune response to pneumococcal serotypes. However, when Tdap was administered 3–4 weeks before PCV13, the immune response to pneumococcal serotypes decreased [18]. The lower immune response to pneumococcal serotypes can be due to immune interference due to similarities in carrier proteins, which is called carrier-induced epitopic suppression (CIES) [10,27]. The PCV13 vaccine is conjugated to CRM197, a non-toxic mutant form of diphtheria toxin [28]. Similarly, MCV4 vaccines are conjugated to either diphtheria toxin (DT) or CRM197. Carrier proteins can trigger anti-carrier antibodies, which may interfere with the immune response to conjugate vaccines, particularly when administered at intervals [10]. The findings support the concomitant administration of PCV13 with Tdap and MCV4 vaccines and suggest that if it is not possible to vaccinate travelers concomitantly, and time allows, the pneumococcal vaccine should be administered first, followed by the Tdap vaccine four weeks later [28,29]. However, evidence on the clinical efficacy of this strategy is limited, and further investigations are needed.

The effect of PCV13 or MCV4 vaccination on the immune response to Tdap was outside the scope of our systematic review. However, it is worth mentioning that in the study by Tashani et al., PCV13 and MCV4 vaccination prior to Tdap administration demonstrated an enhanced immune response to tetanus and diphtheria [18]. This phenomenon has been reported in some previous studies and is referred to as carrier-specific enhancement of the immune response [10]. The clinical significance of the “carrier-specific enhancement of the immune response” remains unknown.

Immunosenescence or age-associated alterations in the immune system, including lymph nodes, immune cells, and inflammatory factors, can potentially decrease the immune response to vaccines in older individuals [30]. However, the study by Song et al. indicated that the co-administration of the pneumococcal and diphtheria-tetanus vaccines in individuals over 50 was not inferior to the single administration of the pneumococcal vaccine [19].

Many immunocompromised individuals with different degrees of compromised immune function are among travelers each year and are at risk of infectious diseases [31]. We found three studies that included immunocompromised individuals, including patients with rheumatoid arthritis who received baricitinib [20], those with MM [22], and hematopoietic stem cell transplant recipients [23]. The studies showed that pneumococcal vaccines elicited immune responses to most serotypes when co-administered with other vaccines in immunocompromised adults, albeit with some variability. Although these findings are promising, some of the studies are quite small, and there is some degree of risk of bias. Therefore, the findings should be interpreted cautiously, and it is difficult to make evidence-based recommendations for vaccine scheduling in immunocompromised populations. It is necessary to perform further well-designed studies in immunocompromised individuals, and decisions about vaccination in this group should be individualized according to the cause of immunosuppression, immunosuppressive agents they receive, the presence of further treatment plans, and the age group. Furthermore, it is crucial to investigate the effectiveness of vaccines in immunocompromised individuals.

The study by Palazzo et al. is the only one that vaccinated immunocompromised adults with a combination of vaccines, including pneumococcal and a live vaccine, MMR [23]. The co-administration did not impact the immune response to pneumococcal serotypes. Vaccines such as MMR, yellow fever, and oral typhoid vaccines are live-attenuated vaccines, and their administration in immunocompromised individuals has certain limitations.

As mentioned earlier, we could not find any information regarding the co-administration of pneumococcal and yellow fever, Japanese encephalitis, rabies, and typhoid vaccines in adult immunocompetent or immunocompromised individuals. These vaccines are among the common vaccines recommended to travelers to endemic areas; therefore, further research is warranted.

Using a broad combination of search terms, including a wide range of study types, reviewing reference lists of relevant articles, and searching multiple databases by two authors independently were the strengths of this systematic review. However, there were some limitations. We excluded articles published in journals not indexed in the selected databases and articles published in languages other than English. Additionally, expert opinions or conference abstracts were not included, and it is possible that some information from the gray literature was missed. Other than the limited number of studies (six studies), the populations, vaccine types, and co-administration protocols vary across the included studies, which may impact the external validity of the findings, and it was not possible to perform a meta-analysis. Furthermore, while many cited studies investigate the immunogenicity and safety of specific vaccines, there are limitations in directly applying these findings to all travel-related vaccine combinations. While our primary focus was on the impact of co-administered vaccines on pneumococcal immunogenicity, the reverse effect, i.e., the impact of pneumococcal vaccines on the immunogenicity of other vaccines, remains an important research question. Furthermore, all included studies used PCV13; however, none investigated the impact of polysaccharide vaccines or newer conjugate vaccines such as PCV15, PCV20, or PCV21. Moreover, many of the studies did not have a control group with single-PCV administration. Future studies should investigate these newer vaccines, as they are increasingly recommended for adult immunization.

## 5. Conclusions

In conclusion, this review highlights the safety and immunogenicity of pneumococcal vaccines in both immunocompetent and immunocompromised adults when co-administered with common travel vaccines. The focus of our systematic review was on vaccines recommended for travelers. However, many of these vaccine combinations also have implications for general routine immunization practices beyond the context of travel preparation. While co-administration strategies appear generally safe, the timing of vaccine administration may play a role in maximizing immunogenicity, particularly in immunocompromised populations. Our finding highlights an important knowledge gap and underscore the need for further high-quality research. Further research can refine vaccine schedules and ensure optimal protection in individuals with varying degrees of immune suppression.

## Figures and Tables

**Figure 1 vaccines-13-00643-f001:**
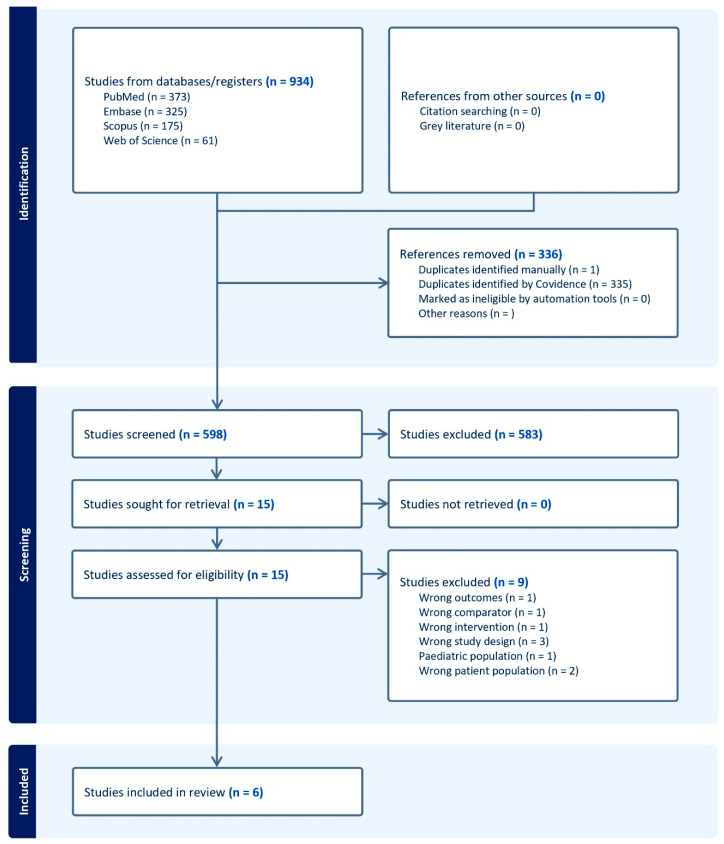
PRISMA flowchart for included studies. Immunogenicity and adverse effects of pneumococcal vaccines co-administered with hepatitis A, hepatitis B, yellow fever, diphtheria-tetanus, Japanese encephalitis, rabies, typhoid, or meningococcal vaccine: A systematic review.

**Figure 2 vaccines-13-00643-f002:**
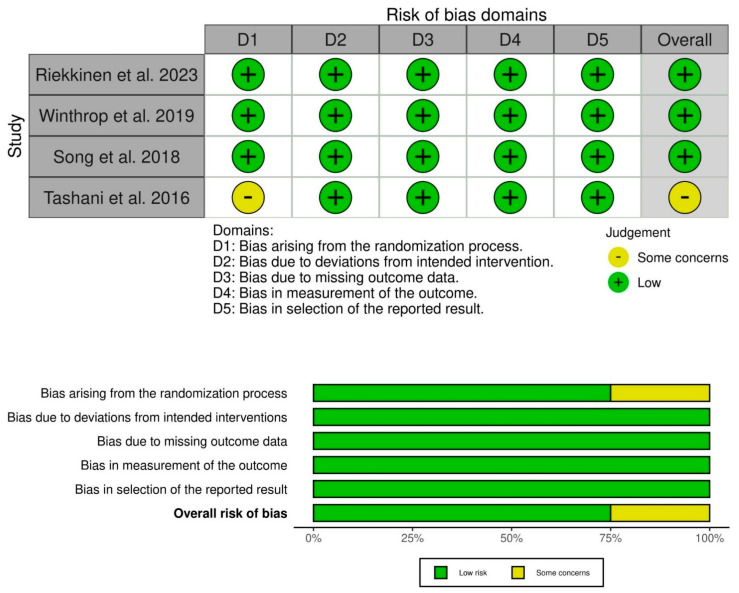
Risk of bias in randomized clinical trials (RCTs). Three of the four RCTs had a low risk of bias, and one had some concerns in the randomization process [17,18,19,20].

**Figure 3 vaccines-13-00643-f003:**
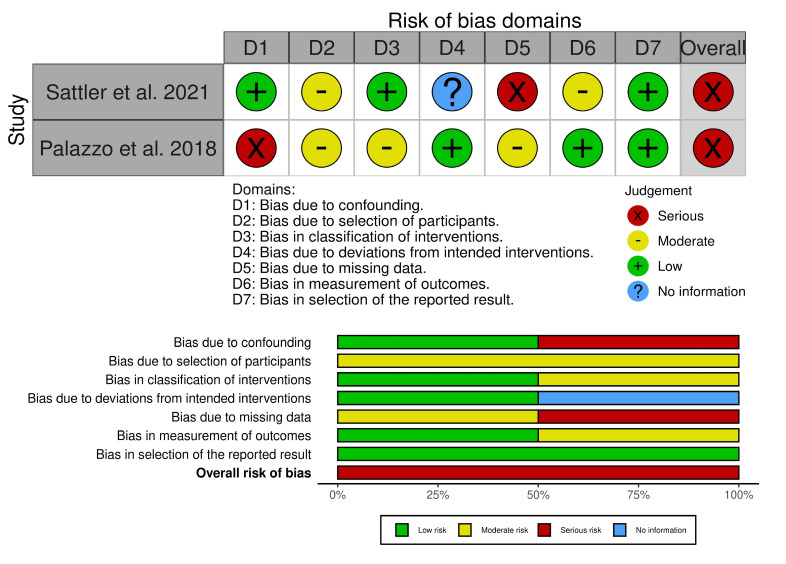
Risk of bias for non-randomized studies. The two non-randomized studies had a serious risk of bias, with specific concerns about missing data and confounding factors [22,23].

**Table 1 vaccines-13-00643-t001:** Characteristics of the included studies.

First Author/Publication Year/Country in Which the Study Was Conducted	Study Design	Pneumococcal Vaccine (Producer)	Population Description	Contribution to Review	Limitations
**Studies in Immunocompetent Adults**					
Riekkinen/2023/Finland and Sweden	RCT	Prevnar13^®^ (Pfizer Inc., New York, NY, USA)	Three groups of 304 healthy adults were randomized in a 1:1:1 ratio: HepA group: 102 participants, mean age 47 years (SD: 18), 30 males (38%); PCV + HepA group: 101 participants, mean age 44.6 years (SD: 18.5), 34 males (40%); PCV group: 101 participants, mean age 46.9 years (SD: 18.4), 28 males (32.2%).	Explores the impact of coadministration of pneumococcal conjugate vaccine with Hepatitis A vaccine, highlighting potential immune interference. Directly relevant to coadministration concerns in travel vaccines.	Absence of long-term follow-up
Tashani/2016/Australia	RCT	Prevenar13^®^ (Pfizer, Inc.)	A total of 111 adults were randomized in a 1:1:1 ratio into three groups: Tdap → MCV4 + PCV13 group: 37 participants, mean age 39 years (SD: 11), 18 males (51%); Tdap + MCV4 + PCV13 group: 37 participants, mean age 42.4 years (SD: 12), 15 males (43%); MCV4 + PCV13 → Tdap group: 37 participants, mean age 42 years (SD: 12), 14 males (38%).	Assesses the effects of administering Tdap before, simultaneously with, or after pneumococcal vaccines. Addresses timing and immune response, key for travel vaccine protocols.	The small sample size limited the analysis, and the effect of MCV4 was difficult to isolate since PCV13 was not administered alone.
Song/2018/South Korea	RCT	Prevnar 13^®^ (No information)	A total of 462 subjects aged ≥50 years were randomized in a 1:1:1 ratio: PCV + Td group: 154 participants, mean age 58 years (range: 57–58), 41 males (27%); PCV group: 154 participants, mean age 58 years (range: 57–59), 37 males (25%); Td group: 154 participants, mean age 58 years (range: 57–59), 36 males (24%).	Evaluate immunogenicity and safety of coadministered tetanus-diphtheria and pneumococcal conjugate vaccines in older adults. Relevant for understanding the immune response when combining these vaccines.	Only a limited number of pneumococcal antigens could be tested (4 out of 13), and there was insufficient information on prior Td vaccination.
**Studies in Immunocompromised Patients**					
Winthrop/2019/USA	RCT	Prevnar 13^®^ (Pfizer, Inc.)	A total of 106 patients with rheumatoid arthritis were receiving baricitinib. The mean age was 55 years (SD: 12), and 21 patients (20%) were male.	Study vaccine responses in patients with rheumatoid arthritis receiving baricitinib. Relevant for understanding pneumococcal vaccine efficacy in immunosuppressed adults.	No control group was available to evaluate the effect of MTX on vaccine response.
Sattler/2020/Germany	Retrospective cohort study	Prevenar13^®^ (Pfizer, Inc.); Synflorix^®^ (GlaxoSmithKline)	Sixty-seven alloHSCT recipients received PCV vaccines (55 received PCV13 and 12 received PCV10) concomitantly with tetanus, diphtheria, pertussis, *Haemophilus influenzae*, and poliomyelitis vaccines. The mean age was 52 years (19–71), and 36 participants (54%) were male.	Analyzes primary vaccination in adults post-allogeneic hematopoietic stem cell transplantation, including pneumococcal vaccines. Highlights safety and efficacy in a vulnerable population.	There was a large time gap between vaccination and the measurement of the immune response.
Palazzo/USA/2017	Retrospective cohort study	Prevnar 13^®^ (Pfizer, Inc.)	A total of 122 multiple myeloma (MM) patients following autologous hematopoietic cell transplantation (aHCT) were assigned to two groups: Lenalidomide group: 91 patients, mean age 58 years (range: 42–75), 49 males (54%); No lenalidomide group: 31 patients, mean age 57 years (range: 38–71), 15 males (48%). All patients received the following vaccines: *Haemophilus influenzae* type B (HiB) conjugate vaccine, inactivated poliovirus vaccine, PCV13, Tdap, hepatitis A, and hepatitis B vaccines.	Focuses on the immunogenicity and safety of pneumococcal vaccination in patients undergoing hematopoietic stem cell transplantation. Relevant in establishing pneumococcal vaccine response in immunocompromised adults	Fewer patients were not on lenalidomide (LM) at the time of vaccination.

Abbreviations: aHCT—autologous hematopoietic cell transplantation; alloHSCT—allogeneic hematopoietic stem cell transplantation; HepA—hepatitis A vaccine; HiB—*Haemophilus influenzae* type B conjugate vaccine; MCV4—meningococcal conjugate vaccine quadrivalent; MTX—methotrexate; PCV—pneumococcal conjugate vaccine; PCV10—pneumococcal conjugate vaccine 10-valent; PCV13—pneumococcal conjugate vaccine 13-valent; RCT—randomized controlled trial; Td—tetanus and diphtheria vaccine; Tdap—tetanus, diphtheria, and pertussis vaccine.

## Data Availability

The extracted data is reported here, and further details are available upon request from the corresponding authors.

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
