# Peer review of "Immunogenicity and Safety of Pneumococcal Vaccines Co-Administered with Common Travel Vaccines in Adults: A Systematic Reviewâ€"

_vaccines, 2025, doi:10.3390/vaccines13060643_

Round 1
Reviewer 1 Report
Comments and Suggestions for Authors
The review is well written, but the included studies is 6, which is very limited. I would recommend to add more studies, because as a review article, the readers need more input information, which should be addressed. Overall, the study is fine and cover important point to discuss.
- Input the text in the introduction section: Despite widespread use of co-administration in pediatric populations, there is a notable lack of data on the safety and immunogenicity of pneumococcal vaccine co-administration with travel vaccines in adults.
- Are there key differences in immune response between adults and children that justify the need for separate adult-focused data?
- Why were SARS-CoV-2, Influenza, and Herpes Zoster excluded, even though they are highly relevant to older travelers? Could a brief summary of findings from the prior reviews on these vaccines help contextualize their exclusion?
- Line 93, Why data collected from more than 18 year of age. The persons get immunity in that age, so it would be better to not select the age bar. Please justify!!
- PRISMA flow diagram (which may be included elsewhere, but isn’t referenced here), Please cite it or justify it !!
- Under section 2.4 …. How many of the included RCTs were rated as low risk of bias? How many observational studies had a high risk?
- Did studies at high risk of bias affect your conclusions or meta-analysis results?
- Which domains were most problematic across studies—blinding, selection bias, or selective reporting? .. It would be better to add, if possible!!
- This could lead to concerns about the strength and generalizability of conclusions, especially in a systematic review. Only six studies were included. Was this due to a lack of available data, or did most studies fail to meet quality/inclusion standards? Please clarify!!
- Section 3.2: Winthrop et al. did not include a group receiving PCV13 alone, making it impossible to isolate the effect of co-administration with Tdap on immunogenicity or adverse events. Whereas, Sattler et al. involved multiple vaccines (including PCV13 and Synflorix) administered concomitantly in a highly vulnerable group (post-HSCT), but lacked stratified immunogenicity data or clear controls.
Without this comparison, causality and additive effects of co-administered vaccines can't be determined, undermining the ability to make evidence-based recommendations for vaccine scheduling in immunocompromised populations.
- Studies inclued in the review is 6, which is very limited.. So please justify it
- Can you elaborate on the clinical implications of the observed enhancement of serotype 3 and 4 responses with co-administration of PCV13 and hepatitis A vaccine?
- In cases where simultaneous vaccination isn’t feasible, should pneumococcal vaccines always be prioritized before administering other conjugate vaccines?
Author Response
We would like to thank you for your comments and the opportunity to revise and resubmit.
Please find below our detailed point-by-point responses.
Reviewer 1, Comment 1:
The review is well written, but the included studies is 6, which is very limited. I would recommend to add more studies, because as a review article, the readers need more input information, which should be addressed. Overall, the study is fine and cover important point to discuss.
Response to Reviewer 1, Comment 1:
Thank you for your positive feedback and valuable comment. As this is a systematic review, we strictly followed the PICOS framework and predefined eligibility criteria to ensure methodological rigor and relevance. The limited number of included studies reflects the current state of the evidence on this topic. We believe this finding highlights an important knowledge gap and underscores the need for further high-quality research. We have mentioned this point in the limitations, discussion section, on page 14, lines 437-439, which reads:
“Other than the limited number of studies (six studies), the populations, vaccine types, and co-administration protocols vary across the included studies, which may impact the ex-ternal validity of the findings, and it was not possible to perform a meta-analysis.”
Moreover, we elaborated on page 14, lines 458-461, which reads:
“Our finding highlights an important knowledge gap and underscores the need for further high-quality research. Further research can refine vaccine schedules and ensure optimal protection in individuals with varying degrees of immune suppression.”
Reviewer 1, Comment 2:
Input the text in the introduction section: Despite widespread use of co-administration in pediatric populations, there is a notable lack of data on the safety and immunogenicity of pneumococcal vaccine co-administration with travel vaccines in adults. Are there key differences in immune response between adults and children that justify the need for separate adult-focused data?
Response to Reviewer 1, Comment 2:
Thank you for this insightful suggestion. We agree that highlighting the contrast between paediatric and adult data strengthens the rationale for our review. In response, we elaborated on Page 2, lines 58-74, which reads:
“Vaccine co-administration, defined as administering two or more vaccines in the same visit, is a routine clinical practice aiming to increase vaccination coverage [8]. However, our knowledge of the immunogenicity or safety of the co-administered vaccines in adults is limited since most evidence comes from studies in children [8]. The gap in evidence in adults is especially important because immune responses in adults can differ from those in children due to several factors, including immunosenescence, prior antigen exposure, and the higher prevalence of chronic comorbidities [9]. These age-related variations can influence both the magnitude and durability of vaccine-induced protection, suggesting that findings from paediatric studies cannot be directly extrapolated to adult populations. In addition, co-administration of vaccines can result in potential immuno-logical interactions such as carrier-induced epitopic suppression (CIES), bystander interference, and carrier specific enhancement of T-Cell Help which can potentially affect immune response to vaccines [10]. Therefore, we conducted a systematic review to answer whether the co-administration of pneumococcal vaccines with common travel vaccines in adults impacts the immunogenicity and safety of the administered pneumococcal vaccine in adults. The findings can help to decrease knowledge gaps, inform healthcare practitioners, and refine vaccination schedules.”
Reviewer 1, Comment 3:
Why were SARS-CoV-2, Influenza, and Herpes Zoster excluded, even though they are highly relevant to older travelers? Could a brief summary of findings from the prior reviews on these vaccines help contextualize their exclusion?
Response to Reviewer 1, Comment 3:
Thank you for your comment. We have addressed those infections in two separate systematic review and meta-analyses, as was mentioned on page 2, lines 85–88, which reads:
“However, we have investigated the co-administration of pneumococcal vaccines with these vaccines in two previous systematic reviews and meta-analyses, and therefore, SARS-CoV-2, Influenza, and Herpes Zoster were excluded from the current work [12,13].”
Reviewer 1, Comment 4:
Line 93, Why data collected from more than 18 year of age. The persons get immunity in that age, so it would be better to not select the age bar. Please justify!!
Response to Reviewer 1, Comment 3:
Thank you for your comment. We included only individuals aged 18 years and older because this age is widely recognized as the standard definition of adulthood in clinical research and public health guidelines.
Reviewer 1, Comment 5:
PRISMA flow diagram (which may be included elsewhere, but isn’t referenced here), Please cite it or justify it !!
Response to Reviewer 1, Comment 5:
Please see Figure 1, the PRISMA flow diagram.
Reviewer 1, Comment 6:
Under section 2.4 …. How many of the included RCTs were rated as low risk of bias? How many observational studies had a high risk? Did studies at high risk of bias affect your conclusions or meta-analysis results? Which domains were most problematic across studies—blinding, selection bias, or selective reporting? .. It would be better to add, if possible!!
Response to Reviewer 1, Comment 6:
Thank you for your comment. Your question is addressed on page 10, section 3.3 Risk of Bias, which reads:
“Three of the four RCTs had a low risk of bias, and one had some concerns regarding the randomization process (Figure 2). However, both non-randomized studies had a serious risk of bias, with specific concerns related to missing data and confounding factors (Figure 3).”
Furthermore, we were mentioned on page 4, section 2.5, Synthesis of Results:
“Due to the heterogeneity in inclusion criteria, populations, and vaccine combinations across the studies, performing a meta-analysis was not possible. Therefore, the results were reported narratively.”
Therefore, our conclusions were based on a narrative synthesis. We elaborated and included this point to the limitations of the study (page 14, line 437-440).
Reviewer 1, Comment 7:
This could lead to concerns about the strength and generalizability of conclusions, especially in a systematic review. Only six studies were included. Was this due to a lack of available data, or did most studies fail to meet quality/inclusion standards? Please clarify!!
Response to Reviewer 1, Comment 7:
Please see the Response to Reviewer 1, Comment 1:
Reviewer 1, Comment 8:
Section 3.2: Winthrop et al. did not include a group receiving PCV13 alone, making it impossible to isolate the effect of co-administration with Tdap on immunogenicity or adverse events. Whereas, Sattler et al. involved multiple vaccines (including PCV13 and Synflorix) administered concomitantly in a highly vulnerable group (post-HSCT), but lacked stratified immunogenicity data or clear controls.
Without this comparison, causality and additive effects of co-administered vaccines can't be determined, undermining the ability to make evidence-based recommendations for vaccine scheduling in immunocompromised populations.
Response to Reviewer 1, Comment 8:
Thank you for your valuable comment. We fully agree with your comment and have acknowledged these limitations in the respective parts (on page 9, section 3.2). We elaborated on page 13, lines 411-419, which reads:
“Although these findings are promising, some of the studies are quite small, and there are some degrees of risk of bias. Therefore, the findings should be interpreted cautiously, and it is difficult to make evidence-based recommendations for vaccine scheduling in immunocompromised populations. It is necessary to perform further well-designed studies in immunocompromised individuals, and decisions about vaccination in this group should be individualized according to the cause of immunosuppression, immunosuppressive agents they receive, the presence of further treatment plans, and the age group. Further-more, it is crucial to investigate the effectiveness of vaccines in immunocompromised individuals.”
Reviewer 1, Comment 9:
Studies included in the review is 6, which is very limited.. So please justify it
Response to Reviewer 1, Comment 9:
Please see the Response to Reviewer 1, Comment 1:
Reviewer 1, Comment 10:
Can you elaborate on the clinical implications of the observed enhancement of serotype 3 and 4 responses with co-administration of PCV13 and hepatitis A vaccine?
Response to Reviewer 1, Comment 11:
Thank you for your insightful question. We have discussed this point in the discussion section on page 12-13, lines 370-375, which reads:
“Both serotypes 3 and 4 are among the common causes of invasive pneumococcal infections in adults [24]. Serotype 3 is recognized as one of the most problematic pneumococcal serotypes, characterized by poor or non-antibody responses after vaccination and causing severe infections such as empyema, meningitis, and sepsis [25]. Therefore, a strategy that can improve the immune response against this serotype is of clinical value and can be investigated in future studies.”
Reviewer 1, Comment 12:
In cases where simultaneous vaccination isn’t feasible, should pneumococcal vaccines always be prioritized before administering other conjugate vaccines?
Response to Reviewer 1, Comment 12:
Thank you for highlighting this point. We addressed your point on page 13, lined 386-391, which reads:
“The findings support the concomitant administration of PCV13 with Tdap and MCV4 vaccines and suggest that if it is not possible to vaccinate travellers concomitantly, and time allows, the pneumococcal vaccine should be administered first, followed by the Tdap vaccine four weeks later [27,28]. However, evidence on the clinical efficacy of this strategy is limited, and further investigation is needed.”
Reviewer 2 Report
Comments and Suggestions for Authors
The authors have written a well-structured and timely systematic review on the immunogenicity and safety of pneumococcal vaccines co-administered with common travel vaccines in adults. This is a clinically relevant and under-explored area, and Niyati and colleagues make a valuable contribution to the field of travel medicine and adult immunisation practices.
The methodology is solid. The search strategy is clear, inclusion criteria are appropriate, and the authors provide a transparent assessment of the limitations in the literature, particularly the small number of available studies and the presence of bias in some.
I suggest the authors to describe slightly on the mechanism of carrier-induced epitopic suppression (CIES) earlier in the Introduction or Methods, as it becomes a central theme in the Discussion. If available, clarify whether any included studies reported on the timing intervals of vaccine co-administration, especially where immunogenicity was found to differ.
In addition, check that all vaccine brand names and acronyms (e.g., PCV13, Tdap, MCV4) are defined on first use and applied consistently throughout.
Regarding the figures, i suggest:
- Simplify long titles, in Table 1 “How does the study contribute to this systematic review?” could be “Contribution to Review”.
- Verify, if appropriate, the use of subheadings to separate immunocompetent from immunocompromised populations within the same table; expecially i suggest, in popolation description in Table 1 .
- Improve image resolution of Figure 1.
While the overall manuscript is clear, there are part of paper as in the Results and Discussion that could be tightened. Some sentences feel repetitive, and simplifying the structure could help the key findings stand out more clearly. I'd also recommend a light language editing.
Author Response
We would like to thank you for your comments and the opportunity to revise and resubmit.
Please find below our detailed point-by-point responses.
Reviewer 2
The authors have written a well-structured and timely systematic review on the immunogenicity and safety of pneumococcal vaccines co-administered with common travel vaccines in adults. This is a clinically relevant and under-explored area, and Niyati and colleagues make a valuable contribution to the field of travel medicine and adult immunisation practices. The methodology is solid. The search strategy is clear, inclusion criteria are appropriate, and the authors provide a transparent assessment of the limitations in the literature, particularly the small number of available studies and the presence of bias in some.
Reviewer 2, Comment 1:
I suggest the authors to describe slightly on the mechanism of carrier-induced epitopic suppression (CIES) earlier in the Introduction or Methods, as it becomes a central theme in the Discussion.
Response to Reviewer 2, Comment 1:
Thank you for your comment. We elaborated in the introduction, on page 2, lines 64-74, which reads:
“These age-related variations can influence both the magnitude and durability of vaccine-induced protection, suggesting that findings from paediatric studies cannot be directly extrapolated to adult populations. In addition, co-administration of vaccines can result in potential immuno-logical interactions such as carrier-induced epitopic suppression (CIES), bystander interference, and carrier specific enhancement of T-Cell Help which can potentially affect immune response to vaccines [10]. Therefore, we conducted a systematic review to answer whether the co-administration of pneumococcal vaccines with common travel vaccines in adults impacts the immunogenicity and safety of the administered pneumococcal vaccine in adults. The findings can help to decrease knowledge gaps, inform healthcare practitioners, and refine vaccination schedules.”
Reviewer 2, Comment 2:
If available, clarify whether any included studies reported on the timing intervals of vaccine co-administration, especially where immunogenicity was found to differ.
Response to Reviewer 2, Comment 2:
Thank you for your comment. None of the included studies reported on varied timing intervals between vaccine administrations; all assessed simultaneous co-administration during the same clinical visit. Therefore, it was not possible to evaluate the impact of timing on immunogenicity.
Reviewer 2, Comment 3:
In addition, check that all vaccine brand names and acronyms (e.g., PCV13, Tdap, MCV4) are defined on first use and applied consistently throughout.
Response to Reviewer 2, Comment 3:
Thank you for your helpful suggestion. We have applied your point through the manuscript.
Reviewer 2, Comment 4:
Regarding the figures, i suggest:
Simplify long titles, in Table 1 “How does the study contribute to this systematic review?” could be “Contribution to Review”.
Response to Reviewer 2, Comment 4:
Thank you for your comment. It has been revised.
Reviewer 2, Comment 5:
Verify, if appropriate, the use of subheadings to separate immunocompetent from immunocompromised populations within the same table; expecially i suggest, in popolation description in Table 1 .
Response to Reviewer 2, Comment 5:
Thank you for your comment. We have added two subheadings in Table 1, dividing the studies into those conducted in immunocompetent and immunocompromised populations.
Reviewer 2, Comment 6:
Improve image resolution of Figure 1.
Response to Reviewer 2, Comment 6:
The high-resolution image has been attached.
Reviewer 2, Comment 7:
Comments on the Quality of English Language
While the overall manuscript is clear, there are part of paper as in the Results and Discussion that could be tightened. Some sentences feel repetitive, and simplifying the structure could help the key findings stand out more clearly. I'd also recommend a light language editing.
Response to Reviewer 2, Comment 7:
We have requested for English language edit as requested.

Reviewer 3 Report
Comments and Suggestions for Authors
Summary
The authors point out that co-administered vaccines can affect immune responses and safety. Their aim was to conduct a systematic literature review on the safety and immunogenicity of pneumococcal vaccine administered with vaccines commonly administered for travel. Using the PRISMA guidelines and the PICOS process for keyword selection, they searched several reported databases covering the range of study designs and publications related to the subject.
They report that, out of 598 publications, six were included in their analysis; three from immunocompetent subjects and six from immunocompromised subjects. They report that Tdap and Hepatitis A vaccines administered to immunocompetent subjects were safe and immunogenic; and that similar findings were observed for immunocompromised subjects co-administered Tdap, Hepatitis A or Hepatitis B vaccines. They report that no studies investigated co-administration with yellow fever, rabies, Japanese encephalitis, and typhoid vaccines; and further that two non-randomised studies in immunocompromised individuals had a substantial risk of bias.
They conclude that pneumococcal vaccine co-administered with Tdap and Hepatitis A and Tdap vaccines is safe and immunogenic in both immunocompetent and immunocompromised persons. They also report that there is a knowledge gap and that more high quality studies are needed.
General comments
The search strategy is well-described and detailed in the methods section, as is the authors’ assessment of the risk of bias in both RCTs and non-randomised studies, as well as was the categorisation of the risk of bias. The search strategy section appears to be exhaustive. The authors explain their reasoning for not conducting a meta-analysis.
The results section lists the findings by study and group, are exhaustively covered, and provide a clear explanation of the study limitations for each study. The risks of bias for the studies are explained, along with a figure (Figure 2) showing the risk domains and levels of concern for randomised studies, and a separate figure (Figure 3) for non-randomised studies.
In the discussion section, both the general and the serotype-specific effects of co-administration are explored and discussed, in addition to the time interval between vaccinations. The authors also discuss the effect of the carrier proteins on responsiveness.
The authors also acknowledge the relatively small sample sizes for the studies that included immunocompromised subjects. They also address the absence of studies involving specific travel-related vaccines.
The study limitations appear to be complete and thorough, and the conclusions sound.
Specific comments
None.
Author Response
We would like to thank you for your comments and the opportunity to revise and resubmit.
Please find below our detailed point-by-point responses.
Reviewer 3
The authors point out that co-administered vaccines can affect immune responses and safety. Their aim was to conduct a systematic literature review on the safety and immunogenicity of pneumococcal vaccine administered with vaccines commonly administered for travel. Using the PRISMA guidelines and the PICOS process for keyword selection, they searched several reported databases covering the range of study designs and publications related to the subject.
They report that, out of 598 publications, six were included in their analysis; three from immunocompetent subjects and six from immunocompromised subjects. They report that Tdap and Hepatitis A vaccines administered to immunocompetent subjects were safe and immunogenic; and that similar findings were observed for immunocompromised subjects co-administered Tdap, Hepatitis A or Hepatitis B vaccines. They report that no studies investigated co-administration with yellow fever, rabies, Japanese encephalitis, and typhoid vaccines; and further that two non-randomised studies in immunocompromised individuals had a substantial risk of bias.
They conclude that pneumococcal vaccine co-administered with Tdap and Hepatitis A and Tdap vaccines is safe and immunogenic in both immunocompetent and immunocompromised persons. They also report that there is a knowledge gap and that more high quality studies are needed.
General comments
Reviewer 3, Comment 1:
The search strategy is well-described and detailed in the methods section, as is the authors’ assessment of the risk of bias in both RCTs and non-randomised studies, as well as was the categorisation of the risk of bias. The search strategy section appears to be exhaustive. The authors explain their reasoning for not conducting a meta-analysis.
Response to Reviewer 3, Comment 1:
Thank you for your comment, as mentioned on page 4, section 2.5, Synthesis of Results:
“Due to the heterogeneity in inclusion criteria, populations, and vaccine combinations across the studies, performing a meta-analysis was not possible. Therefore, the results were reported narratively.”
Reviewer 3, Comment 2:
The results section lists the findings by study and group, are exhaustively covered, and provide a clear explanation of the study limitations for each study. The risks of bias for the studies are explained, along with a figure (Figure 2) showing the risk domains and levels of concern for randomised studies, and a separate figure (Figure 3) for non-randomised studies.
In the discussion section, both the general and the serotype-specific effects of co-administration are explored and discussed, in addition to the time interval between vaccinations. The authors also discuss the effect of the carrier proteins on responsiveness.
The authors also acknowledge the relatively small sample sizes for the studies that included immunocompromised subjects. They also address the absence of studies involving specific travel-related vaccines. The study limitations appear to be complete and thorough, and the conclusions sound.
Response to Reviewer 3, Comment 2:
We appreciate your positive feedback.